# Short-Term l-arginine Treatment Mitigates Early Damage of Dermal Collagen Induced by Diabetes

**DOI:** 10.3390/bioengineering11040407

**Published:** 2024-04-21

**Authors:** Irena Miler, Mihailo D. Rabasovic, Sonja Askrabic, Andreas Stylianou, Bato Korac, Aleksandra Korac

**Affiliations:** 1Center for Biosystems, BioSense Institute, University of Novi Sad, Dr Zorana Djindjica 1, 21000 Novi Sad, Serbia; irena.miler@biosense.rs; 2Institute of Physics Belgrade, National Institute of the Republic of Serbia, University of Belgrade, Pregrevica 118, 11000 Belgrade, Serbia; sonask@ipb.ac.rs (S.A.); 3School of Science, European University Cyprus, 6 Diogenous Str., Egkomi, Nicosia 2404, Cyprus; stylianou.c.andreas@ucy.ac.cy; 4Institute for Biological Research “Sinisa Stankovic”, National Institute of the Republic of Serbia, University of Belgrade, Bulevar Despota Stefana 142, 11000 Belgrade, Serbia; koracb@ibiss.bg.ac.rs; 5Center for Electron Microscopy, Faculty of Biology, University of Belgrade, Studentski trg 16, 11000 Belgrade, Serbia

**Keywords:** dermal collagen I, diabetes, l-arginine, polarization-resolved SHG imaging, Raman spectromicroscopy, AFM

## Abstract

Changes in the structural properties of the skin due to collagen alterations are an important factor in diabetic skin complications. Using a combination of photonic methods as an optic diagnostic tool, we investigated the structural alteration in rat dermal collagen I in diabetes, and after short-term l-arginine treatment. The multiplex approach shows that in the early phase of diabetes, collagen fibers are partially damaged, resulting in the heterogeneity of fibers, e.g., “patchy patterns” of highly ordered/disordered fibers, while l-arginine treatment counteracts to some extent the conformational changes in collagen-induced by diabetes and mitigates the damage. Raman spectroscopy shows intense collagen conformational changes via amides I and II in diabetes, suggesting that diabetes-induced structural changes in collagen originate predominantly from individual collagen molecules rather than supramolecular structures. There is a clear increase in the amounts of newly synthesized proline and hydroxyproline after treatment with l-arginine, reflecting the changed collagen content. This suggests that it might be useful for treating and stopping collagen damage early on in diabetic skin. Our results demonstrate that l-arginine attenuates the early collagen I alteration caused by diabetes and that it could be used to treat and prevent collagen damage in diabetic skin at a very early stage.

## 1. Introduction

Diabetes is a complex metabolic disorder caused by a lack of insulin synthesis, secretion, or action. According to the International Diabetes Federation, diabetes is one of the most widespread diseases, causing over one million deaths each year [1]. Early diagnosis and treatment, along with improved patient control, can reduce complications and offer the possibility of reversing the disease if recognized at an early preclinical stage.

In diabetes, a high blood glucose level (i.e., hyperglycemia) causes tissue damage, disrupts organ homeostasis, and leads to serious medical complications. Despite their significant contribution to diabetes, pathological skin changes often go underdiagnosed and neglected. Hyperglycemia causes changes in biochemical processes in the skin that lead to an increased level of reactive oxygen species (ROS) and a decreased synthesis of nitric oxide (NO) [2]. As a result, redox imbalances cause metabolic abnormalities that change the skin’s functional and mechanical properties. These include changes in blood circulation (microcirculation), which cause vascular disorders, and problems with the nervous and somatosensory systems, leading to different types of neuropathies [3]. In addition, NO deficiency leads to poor skin nutrition and thermoregulation, reduced regenerative capacity, infection, impaired wound healing, and the development of skin ulcers in the diabetic state [4,5]. Overall, diabetic complications lead to an altered composition of the extracellular matrix, inflammation, the apoptosis of endothelial cells, inhibited keratinocyte proliferation and migration, protein biosynthesis, impaired phagocytosis and chemotaxis of various cells, fibroblast proliferation, and the reduced number and synthesis of collagen [6,7,8].

Previous studies have shown that the mechanical properties of collagen are altered in diabetes and that the disorganization of collagen fibrils and nanoscale fragmentation lead to the extensive disruption of dermal collagen integrity [9,10]. In a diabetic animal model, we previously published that redox state disturbances (affecting NO) occur in the skin in the early phase of the disease [11]. Impaired NO synthesis in the diabetic state is also a consequence of significantly reduced l-arginine concentrations in plasma and in many tissues [12,13]. The changes in l-arginine/NO homeostasis can be compensated by l-arginine supplementation and, thus, restore NO signaling by increasing arginine levels.

In addition to its effects on endothelial relaxation and vascular dysfunction, the beneficial effects of l-arginine in diabetes lead to an improvement in energy homeostasis throughout the body. Two main effects that have a direct action on the pancreas, i.e., the induction of β-cell regeneration [14] and the regulation of insulin synthesis and secretion [15,16].

The beneficial effects of l-arginine in diabetes have been widely described both in animal and human studies, in vivo and in vitro [13,17]. These studies demonstrated that the intake of l-arginine significantly improves glucose metabolism, insulin resistance, and insulin sensitivity [13,17]. We have previously published that treatment with l-arginine has positive systemic effects. It leads to a decrease in glucose concentration and a restoration of insulin levels in the circulation to the control level [14,18]. Importantly, l-arginine supplementation is considered safe and significant in preventing and delaying tissue damage in diabetes [19]. Therefore, increasing the effectiveness of prevention and treatment of diabetic skin disorders at an early stage reduces the risk of complications and makes it possible to reverse their development.

Advanced biological and medical techniques were used to obtain data for collagen analysis concerning biochemical and structural changes at each hierarchical level. Optical diagnostics, e.g., various types of spectromicroscopy techniques, which provide information on the biomechanical and physiological properties of the skin, are successfully used in clinical practice. In particular, polarization methods are used to assess microstructural changes in the skin caused by cancer, diabetic wounds, burns, neurodegeneration, and age-related and diabetic consequences [9]. A very effective and highly sensitive tool to characterize collagen structure is a polarized second harmonic generation (pSHG) [20]. More recently, Raman spectroscopy has been used as a tool to characterize the components of the extracellular matrix, especially in pathological conditions [21].

It is also important to create techniques for correlative analysis that can detect biochemical changes in collagen at different stages of normal and/or abnormal physiological processes. This is necessary for the creation of a precise therapy. Based on a priori data and supported by information from non-invasive optical imaging techniques, adequate therapy would enable a prompt response and stop the disease’s progression. For the proper correlation of the collagen structure with its mechanical properties, the atomic force microscope (AFM) is the ideal tool. AFM is a unique and powerful method for analyzing the structure and properties of a sample without destroying it [22]. Currently, AFM primarily studies the structure and mechanical properties of collagen-based nanobiomaterials, the interactions between collagen and substrate during the formation of collagen thin films, the collagen–cell interactions, and the collagen–optical radiation interactions [23].

The correlative study of skin aging can use the same skin volume (paraffine sections), as our previous research has demonstrated [24]. In the present study, we integrated a multiplex approach using optical diagnostics to analyze dermal collagen changes in diabetes and after short-term l-arginine treatment.

## 2. Materials and Methods

### 2.1. Experimental Animals and Treatment Protocol

We have used the model of diabetes induction by alloxan and l-arginine treatment previously published by Jankovic et al. [11]. Three-month-old male Mill Hill hybrid hooded rats were divided into two groups: diabetic and non-diabetic. To induce diabetes, after a 12 h fasting period, animals received a single alloxan (Sigma, Darmstadt, Germany) dose of 120 mg (kg body weight^−1^) i.p. The rats with a blood glucose level of ≥12 mMol L^−1^, measured by glucose oxidase reagent strips (GlucoSure test, “Prizma” Kragujevac, Serbia), were considered diabetic. To ensure the validity of the effects of diabetes and/or l-arginine treatments, we have assessed blood glucose levels. The measured blood glucose levels did not differ from previously published data [14]. Both diabetic and non-diabetic groups were additionally separated into two subgroups. One subgroup received l-arginine HCl (2.25%) (Sigma, Germany) in drinking water, while the other untreated subgroup drinking tap water served as a control. The rats were maintained in individual cages with food and drinking water ad libitum. The rats were treated for one week. Treatment of diabetic rats started after diabetes induction. Each experimental group consisted of three animals. The experiments were approved by the Ethics Committee for the Treatment of Experimental Animals of the Institute for Biological Research “Siniša Stanković” in Belgrade and by the Veterinary Directorate of the Ministry of Agriculture and Environmental Protection of the Republic of Serbia.

### 2.2. Skin Sample Preparation, pSHG Imaging, and β-Coefficient Calculation

The rat skin sample preparation and sectioning were performed in the same manner as we previously published (described in detail by Miler et al. [24]). Briefly, parts of the shaved skin from the dorsal side were dissected, rinsed, and fixed in 10% neutral formalin at 4 °C overnight. After the rinsing of fixative, skin samples were dehydrated in a series of ethanol (alcohol) solutions of increasing concentration and routinely embedded in paraffin. The pSHG imaging, β-coefficient calculation, and statistical comparison were also performed as previously published (described in detail by Miler et al. [24]). Only for pSHG images of label-free paraffin sections, there was a slight change regarding the laser power applied. The average laser power in the sample plane was in the range of 13 to 21 mW. From this, we estimated the power densities in the sample to be 13.5 to 22 MW/cm^2^.

### 2.3. Raman Instrumentation and Analysis

Raman spectra of collagen I were obtained from the same paraffin section of the rat skin used for pSHG imaging with the laser beam focused on the collagen fibers. The spectra were acquired using the Xplora Raman system from Horiba Jobin Yvon. The excitation line used was 532 nm, and the monochromator diffraction grating used had 1200 grooves/mm. The power on the sample was kept low to prevent thermal degradation. The spectral wavenumbers were calibrated using the 520.5 cm^−1^ line of silicon. The exposure time was 120 s per position, and spectra were obtained from at least nine different positions on each sample.

Raman spectra were processed in R using the hyperSpec package [Beleites and Sergo [25]]. Processing included first-order polynomial baseline subtraction and normalization according to integral intensity. The low-frequency spectral range (700–1800) cm^−1^ was selected for analysis. Principal component analysis (PCA) was applied to the collagen spectra of control tissue, diabetic tissue, and after l-arginine treatment. PCA score values and mean Raman spectra were used to compare both subgroups.

### 2.4. AFM Instrumentation and Analysis 

AFM assessments were carried out on deparaffinized skin sections using a commercially available AFM system, specifically the Molecular Imaging-Agilent PicoPlus AFM. Silicon nitride cantilevers (MLCT-Bio, cantilever D, Bruker Company, Camarillo, CA, USA) were employed for these measurements. The maximum applied loading force was predetermined at 1.8 nN [26]. AFM measurements were performed by recording 5–10 different 20 × 20 μm^2^ force maps (16 × 16-point grids) per specimen. The collected force maps were analyzed with AtomicJ (https://sourceforge.net/projects/jrobust/, accessed on 12 March 2024) [27] to calculate the sample’s Young’s modulus using the Hertz model, setting the Poisson’s ratio to 0.5.

### 2.5. Picrosirius Red Staining and Image Acquisition

The assessment of collagen levels in tissue sections was conducted using picrosirius red staining (Sigma, Germany). The tissue sections were subjected to deparaffinization, washed with distilled water, and exposed to picrosirius red stain for one hour at room temperature. Subsequently, the sections were rinsed twice with acetic acid, followed by two rinses with absolute ethanol, and mounted in DPX for histological analysis. As a result, Type I collagen fibers, which are thicker, display a red (or orange) color, whereas Type III collagen fibers, which are thinner, appear green or yellow [28,29].

Images of stained tissue sections were captured at 10× magnification using an Olympus BX53 microscope. These images were obtained both without polarizers and with linear polarizers (specifically, the U-POT polarizer, U-ANT analyzer, and U-KPA intermediate attachment, all from Olympus Corp., Tokyo, Japan). Images of the same staining conditions were acquired with identical settings. All image capture processes were carried out digitally using the cellSens acquisition platform from Olympus, Japan.

## 3. Results and Discussion

### 3.1. β-Coefficient Calculation from pSHG Images

For the analysis of the fine structural changes, packing patterns, and 3D organization of collagen I in rat skin during alloxan-induced diabetes and the effect of l-arginine treatment on diabetic skin, we calculated the β-coefficient from pSHG images obtained from the paraffin sections. The β-coefficient describes the “arrangement” of collagen. The coefficient values are in the range of zero to one. The β-coefficient value equal to one means that collagen is perfectly arranged. Contrary to this, the β-coefficient value equal to zero means that collagen is completely disordered. As in our previous studies, we analyzed sections from three animals for each studied group and subgroup (control, treated with l-arginine, diabetic, and diabetic treated with l-arginine). Three different regions of interest (ROI) from each section were randomly selected for imaging. The β-coefficient was calculated for each pixel in the ROI. The average β-coefficient was calculated for each ROI. Results were analyzed after the calculation of the β-coefficient and categorized as either a less, more, or highly ordered structure (Figure 1a inset, β values range from 0/red to 1/green).

Diabetes, as a pathological condition, impairs the skin’s protective mechanism by breaking down the collagen structure [10]. In our model, the control and diabetic groups have a similar β-coefficient (0.16 vs. 0.15 in the control group, Figure 1b), with a tendency towards higher values, possibly due to “patchy” patterns of collagen alteration (area with more ordered and less ordered collagen), suggesting uneven collagen damage along fibers (Figure 1a). A decrease in the β-coefficient in the l-arginine-treated group compared to the control is to be expected, as an important role of l-arginine in skin renewal has been demonstrated [30,31]. Treatment with l-arginine, even if it was for a short duration (7 days), lowered the β-coefficient compared to the group with diabetes, indicating the potential of l-arginine to mitigate or slow down the effects of diabetes on skin collagen. Therefore, we can use it as a premedication for early-stage diabetes.

Since l-arginine improves skin condition by promoting collagen synthesis and cell proliferation [32,33,34] with a short duration of treatment, we were further interested in the biochemical and mechanical changes in collagen and performed Raman and AFM analyses.

### 3.2. Raman Spectroscopy Analysis

Structural and biochemical spectroscopic analyses were used to evaluate the changes in collagen I of rat skin in alloxan-induced diabetes and the effect of a 7-day treatment with l-arginine. The results are summarized in Figure 2 and Figure 3, which show the mean spectra of the four groups of samples. Appendix A contains the positions of the Raman modes and the corresponding assignations. The vibrational mode assignments are based on the experimental and theoretical Raman spectra of collagen and proteins in general, which can be found in the literature (see Appendix A for references). The results are obtained by combining PCA and the analysis of the mean spectra.

According to Martinez et al. [35], three components of collagen that mainly determine the conformational changes in the collagen structure are amides and the amino acids, proline and hydroxyproline. These components are represented by characteristic vibrational bands in Raman spectra. In order to recognize changes in the secondary structure of collagen, changes in the amide I, II, and III bands are usually followed [36].

The Raman peaks associated with the collagen components (highlighted) are located at 823 cm^−1^, 828 cm^−1^, 858 cm^−1^, 917 cm^−1^, 934 cm^−1^, 1249 cm^−1^, 1276 cm^−1^, 1336 cm^−1^, 1544 cm^−1^, 1557 cm^−1^, and 1594 cm^−1^. The intensities of the first group of modes at 1336 cm^−1^, 1544 cm^−1^, 1557 cm^−1^, and 1594 cm^−1^ are highest in the control group and the diabetic group treated with l-arginine, decreased in diabetes and are the lowest after l-arginine treatment. For the second group of modes, located at 823 cm^−1^, 831 cm^−1^, 858 cm^−1^, 917 cm^−1^, 934 cm^−1^, 1249 cm^−1^, and 1276 cm^−1^, the case is slightly different. Their intensities are the highest in the l-arginine-treated group, lower in the control group, even more slightly lower in the l-arginine-treated diabetic group, and the lowest in the diabetic group.

The Raman peak at 1336 cm^−1^ corresponds to the -CH_3_CH_2_ deforming modes of collagen, such as -CH_3_CH_2_ wagging/-CH_2_ scissoring [35,37]. The deformations of CH_3_/CH_2_ are also reflected in the 1320 cm^−1^ band, which represents the twisting of -CH_3_CH_2_ in collagen [37]. The intensity of this band also decreases in diabetic skin and is restored in diabetic skin treated with l-arginine. This is consistent with the behavior of the band at 1336 cm^−1^ and suggests conformational changes associated with molecular reorientation in diabetic skin and the subsequent recovery of collagen fibers after treatment. The bands at 1544 cm^−1^ and 1557 cm^−1^ could correspond to amide II and tryptophan [38,39]. Excitation lines above 400 nm usually do not observe the amide II band, but since rat collagen does not contain tryptophan, these bands most likely belong to amide II. The vibrational mode positioned at 1594 cm^−1^ corresponds to the vibrations of phenylalanine [38,40]. The amide II band usually affects the β-sheets of the protein, and the shift of this band to lower frequencies indicates changes in the secondary structure of the protein, i.e., conformational changes [39]. As we can see from the above analysis, the intensities of this group of modes decrease in the diabetes skin sample compared to the control but increase again and approach the spectra of the control after treatment with l-arginine.

The mode at 823 cm^−1^ is assigned to the C-C stretching of the collagen backbone. The vibrational mode at 831 cm^−1^ has been assigned to the out-of-plane breathing of tyrosine [38,41]. The mode at 858 cm^−1^ may correspond to the expression of the amino acid proline, hydroxyproline, and the in-plane bending of tyrosine, the side-chain vibrations and C-C stretching modes of proline and the hydroxyproline ring [35], and the C-C vibrations of the collagen backbone; thus, it is not possible to interpret its behavior due to several possible assignments. The peak at 917 cm^−1^ corresponds to the C-C stretching of the proline ring [36,42]. The mode at 934 cm^−1^ also shows a lower intensity in the diabetic skin compared to the control [10,43]. The same mode increases in intensity in the diabetic group after l-arginine treatment (shifted to ~937 cm^−1^), demonstrating that l-arginine reverses the changes induced by diabetes. This mode corresponds to the C-C stretching vibration of proline or the collagen backbone [37,42,44,45,46], suggesting intense structural reorganization. The bands at 1249 cm^−1^ and 1276 cm^−1^ are associated with amide III vibrations [37,47]. This result indicates that the second group of modes exhibits a stronger structural change than the first group of modes. The second group of modes’ intensities decrease in the spectra of diabetic skin, but they increase after l-arginine treatment for the diabetic group and are close to the intensities in the spectra of the control group. This is consistent with diabetes pathology, where we expect the most pronounced conformational changes (i.e., degradation), which affect amino acids with the highest content in the collagen structure, such as proline and hydroxyproline, the most. Researchers demonstrate that the expression of aromatic amino acids like tyrosine can destabilize the tertiary structure of proteins, potentially influencing their 3D structural rigidity [44]. This is consistent with our results, as we observed changes at both the biochemical and mechanical levels, leading to a reorganization of the collagen structure.

Tyrosine also exhibits a further band at 951 cm^−1^, which, according to Joodaki et al. [48] is assigned to the vibrations of out-of-plane atoms and whose intensity increases significantly after treatment with l-arginine, indicating a strong pressure on the 3D structure (or tertiary structure) of the collagen molecule (and thus a reorganization).

In the spectral range 1000–1150 cm^−1^, several bands in the spectra of the diabetic skin increase in intensity compared to the control sample, such as 1035 cm^−1^, 1067 cm^−1^, and 1100 cm^−1^, which are associated with phenylalanine and proline [36,37,49]. According to Carcamo et al. [49], proline has a broader band at 1100 cm^−1^, attributed to N-C-H deformation, with increased intensity in the diabetic skin, suggesting the structural degradation of the collagen molecule [5,50,51], which we consider a pathological mechanism. l-arginine treatment does not reverse the intensities of these modes.

The vibrational modes at 1445 cm^−1^ and 1465 cm^−1^ represent the CH_3_CH_2_ deformation of collagen. The modes in the range of 1640–1700 cm^−1^ represent amide I modes. Both mode groups show increased intensities in the diabetic skin and are very close to each other in the control group and the group treated with l-arginine.

The changes observed in the Raman spectra indicate that treatment with l-arginine attenuates the effects of diabetes on skin collagen I. This is concluded from the analysis of the mean spectra, and the PCA analysis of the groups of spectra shown in Appendix A shows the same. Most of the changes in Raman modes that occur in the diabetic skin are diminished after treatment with l-arginine. Based on the results, the spectra of the diabetic group treated with l-arginine are very similar to the spectra of the control sample.

### 3.3. AFM Analysis

Since we know that mechanical properties provide stability and thereby regulate many cellular functions, we used AFM to analyze the mechanical properties of collagen I in the tissue sections (in terms of Young’s modulus values). The results (Figure 4) show that all groups were characterized by a reduced Young’s modulus, but only the group treated with l-arginine showed a statistically significant reduction in stiffness. Compared to the control group (100%), Young’s modulus decreased by 32.8% in the diabetic group, by 44.7% in the diabetic group treated with l-arginine, and by 72.9% after treatment with l-arginine. This is to be expected, as l-arginine is used for skin renewal through the NO mechanism [31,52,53]. The reduced stiffness of the diabetic dermis is probably due to the degradation and fragmentation of collagen fibrils [10]. In this study, l-arginine treatment is shown to have a significant effect on diabetic skin by reducing stiffness by 11.9% when comparing diabetic dermis to l-arginine-treated diabetic dermis, which may be the result of an increasing number of young and newly synthesized fibrils, with the simultaneous degradation of older collagen fibers. Therefore, l-arginine treatment has an effect on the mechanical properties and alterations in collagen I.

The observed decrease in Young’s modulus, attributed to l-arginine supplementation, might initially seem detrimental, considering the mechanical strength provided by Type I collagen. However, in the context of diabetic skin, which is characterized by increased stiffness and susceptibility to damage, a reduction in stiffness could potentially improve skin pliability and resilience. Our confirmation of l-arginine’s predominant influence on Type I collagen is significant, given Type I’s primary role in skin mechanical properties.

We are aware that changes in glycosaminoglycan (GAG) content and other matrix components could affect tissue stiffness. However, our study’s objective was to assess relative stiffness variations between different experimental groups rather than absolute stiffness values. The literature has previously demonstrated and validated the application of AFM in assessing the mechanical properties of fixed tissue specimens [54].

### 3.4. Picrosirius Red Staining

Picrosirius red staining (Figure 5) additionally shows changes in collagen birefringence that support the changes observed by pSHG and are reflected in the β-coefficient and Young’s modulus.

In our study, the application of Picrosirius red staining revealed thick red collagen I fibers, especially when viewed under polarized light (Figure 5). This finding was pivotal in assessing the impact of l-arginine supplementation on the skin’s extracellular matrix, specifically in the context of diabetic skin conditions.

The administration of l-arginine resulted in a discernible reversal of the pathological alterations typically observed in diabetic skin tissue. Specifically, we noted a reduction in the density and disorganization of collagen I fibers, which are common pathological features in diabetic skin. This outcome suggests that l-arginine supplementation may contribute to the normalization of the skin’s structural integrity by modulating the collagen matrix towards a healthier state, akin to that observed in non-diabetic conditions.

Based on these findings, we infer that l-arginine’s beneficial effects extend beyond merely altering mechanical properties, and they also promote a more organized and potentially functional collagen network within the skin. In correlation with our Raman spectroscopy results, we can further conclude that collagen damage in the early phase of diabetes is a consequence of profound conformational changes and that l-arginine treatments cause a reversal. This insight is crucial for understanding the therapeutic potential of l-arginine in improving skin health in diabetic conditions, offering a promising avenue for further research and application.

Our previous results showed that l-arginine effects (primarily at the level of the pancreas but also systemically) are achieved through NO synthesis, and therefore, NO’s increased bioavailability is observed [11,14,55]. For instance, NO is synthesized in cells from l-arginine in an enzymatic reaction catalyzed by one of the three isoforms of nitric oxide synthase (NOS), i.e., neuronal, endothelial, or inducible, whose expression is tissue-specific. Our previous results indicate that nNOS overexpression as a result of l-arginine supplementation plays a beneficial role in the pancreas [14]. Further, as opposed to the reduction in skin eNOS and iNOS levels caused by diabetes, our results reveal the induction of eNOS and iNOS after l-arginine treatment [11]. Therefore, the beneficial effects of l-arginine on dermal collagen I are a consequence of eNOS and iNOS induction.

## 4. Conclusions

Data from the literature has shown that the changes in nanoscale collagen fibril morphology and mechanical properties that contribute to the aged appearance of skin in diabetes may be a consequence of the accumulation of partially fragmented and cross-linked collagen fragments. Our multiplexed approach to optical diagnostics showed that the diabetes-induced partial intramolecular fragmentation and cross-linking of collagen led to changes in the mechanical properties of the dermis, as we demonstrated with AFM. Collagen’s properties are impaired in people with diabetes because of intermolecular cross-linking, lateral packing, and some intramolecular fragmentation. Even short-term treatment with l-arginine tends to reverse this damage. Therefore, we could use l-arginine supplementation to improve or prevent collagen fiber damage in diabetes.

## Figures and Tables

**Figure 1 bioengineering-11-00407-f001:**
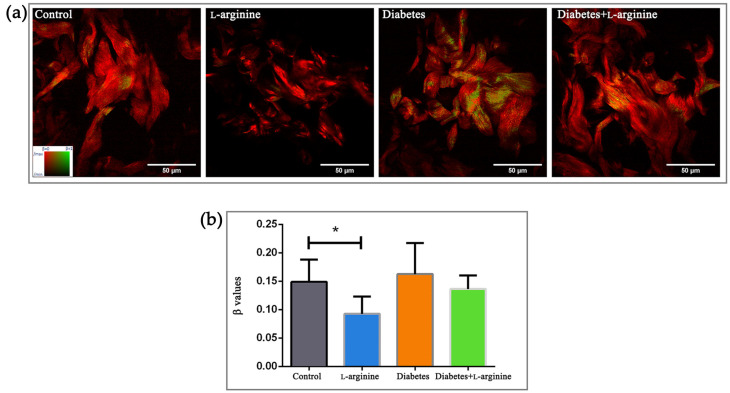
pSHG images of dermal collagen (**a**) and calculated β values (**b**) of control (non-diabetic), l-arginine (l-arginine-treated non-diabetic), diabetic, and diabetic l-arginine-treated rats. Different colors in panel (**a**) indicate the different values of the β-coefficient (as indicated by the color bar in the leftmost panel). Thus, greener images indicate a higher value of the β-coefficient and, therefore, more arranged collagen. The opposite is true for red images. The values represent the mean ± SEM, for each point n = 27, * *p* < 0.05. Scale bars: 50 µm.

**Figure 2 bioengineering-11-00407-f002:**
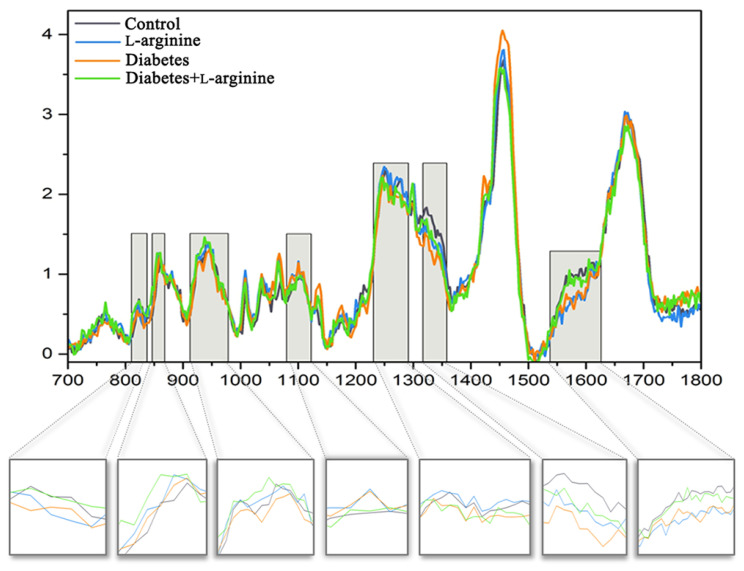
Mean Raman spectra of dermal collagen I structure of control (non-diabetic), l-arginine (l-arginine-treated non-diabetic), diabetic, and diabetic l-arginine-treated rats. The light gray highlighted areas are magnified and present the most intensive changes described in Appendix A.

**Figure 3 bioengineering-11-00407-f003:**
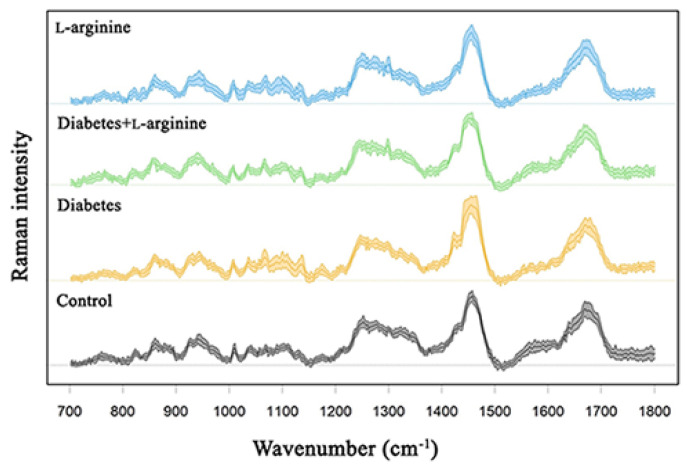
Raman spectra of dermal collagen I of control (non-diabetic), l-arginine (l-arginine−treated non-diabetic), diabetic, and diabetic l-arginine-treated rats.

**Figure 4 bioengineering-11-00407-f004:**
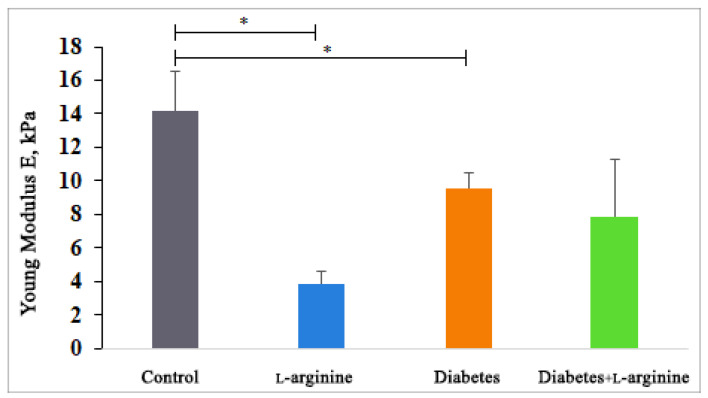
Young’s modulus measured by AFM in control (non-diabetic), l-arginine (l-arginine-treated non-diabetic), diabetic, and diabetic l-arginine-treated rats. The values represent the mean ± SEM, * *p* < 0.05.

**Figure 5 bioengineering-11-00407-f005:**
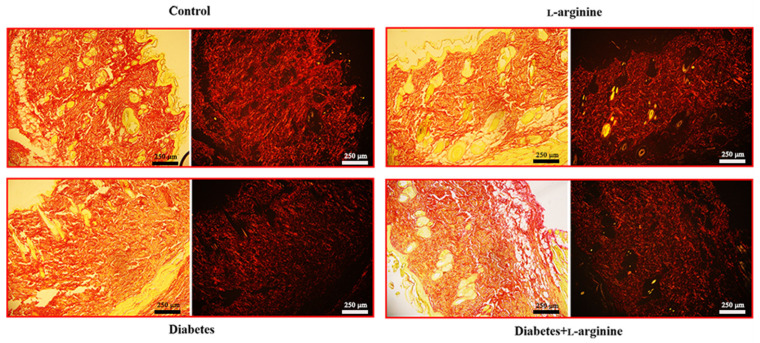
Micrographs of the rat skin of control (non-diabetic), l-arginine (l-arginine-treated non-diabetic), diabetic, and diabetic l-arginine-treated rats stained with picrosirius red and analyzed without polarizers (**left images**) and with linear polarizers (**right images**). Magnification: 10×; original scale bars: 250 µm.

## Data Availability

This article includes all data generated or analyzed during this study.

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
