# Peer review of "Short-Term l-arginine Treatment Mitigates Early Damage of Dermal Collagen Induced by Diabetes"

_bioengineering, 2024, doi:10.3390/bioengineering11040407_

Round 1
Reviewer 1 Report
Comments and Suggestions for Authors
The manuscript by Miler et al. addresses the treatment and prevention of dermal collagen dysfunction through L-arginine dietary supplementation. While I find this manuscript scientifically sound and potentially interesting to readers, I have several questions and concerns that need to be addressed. Firstly, it is necessary to determine whether the subject matter aligns with the scope of the Bioengineering journal (as the scope described on the journal's webpage suggests otherwise), although this decision ultimately rests with the editors. Regarding scientific merits:
1. While the authors demonstrate through several methods the attenuation of collagen ECM damage by oral L-arginine supplementation, they do not attempt to prove the hypothesis that such supplementation increases or restores NO signaling, thereby eliciting regenerative or protective effects on dermal collagen. Although the authors cite three papers (one being a review), they fail to provide primary sources to support this theory. I would therefore recommend a more thorough discussion of the proposed mechanism of L-arginine or the inclusion of analyses demonstrating increased NO or NOS levels in the tissues.
2. It would also be beneficial to present overall changes/markers of the chosen diabetic model and at least correlate the severity of metabolic changes with dermal alterations. Are the differences in blood glucose and subsequent pathological tissue changes the same or at least similar in all diabetic rats (blood glucose levels above ≥12 mMol L-1)? This information could be included in the supplementary material. However, merely stating that the model functions adequately may not suffice for a robust scientific paper.
3. As I am not an expert in the utilization of AFM for measuring tissue mechanical properties, I am concerned about whether the fixation of the skin could influence the measurement outcomes. Although the authors observe significant differences in tissue stiffness between healthy/diabetic and treated rats, could these differences also be affected by variations in the composition and changes in other components of dermal tissue, such as GAG, which yield a less stiff matrix even after standard fixation? Additionally, more details about tissue preparation for AFM and other techniques should be provided. While a previous paper is cited in the methods section (reference 21), this referenced paper also lacks specific information about tissue fixation.
These concerns need to be addressed prior to acceptance for publication.
Reviewer 2 Report
Comments and Suggestions for Authors
The authors evaluated short-term influence of L-arginine on diabetic rat dermis collagen using pSHG, Raman spectra, AFM and Picrosirius red staining. This research is meaningful to diabetic wound healing. However, the description and organization of the results and discussion need to be improved, especially, more details should be given. Moreover, the logic of the statement should be clearer. I suggested that the paper should be double checked carefully and make a major revision before published.
Detailed comments/suggestions:
1. line 108~109, the authors treated the rats using water with L-arginine. Did you try topical treatment using L-arginine?
2. Line 166, an explanation should be given to "β-coefficient".
3. Line 169-170, the full name of the ROIs should be given, the abbreviation should be uniformed, ROIs or ROI?
4. Figure 1, Why did the L-arginine group have different magnification with other groups? The cell seems smaller than others?
5. Figure 1 caption should describe what the red and green color representing. For Fig1b, did the significant differences exist in the other groups?
6. Line 179, the groups in Figure 1 should be explained clearer. Did the control mean non-diabetic rat without treatment? The L-arginine group means non-diabetic rat without treatment?
7. Along with Figure 2, Figure S1 and S2 should be also described in the results and discussion section. You can’t only list the figure in SI without any explanation in the normal manuscript.
8. Line 219-220, what's did the results indicate?
9. Line 235-236, again, what is the meaning of the intensity increase of the group after treated by L-arginine?
10. Line 300-301, Is this negative impact of L-arginine on Young's modulus of collagen good for diabetic skin? Why did you confirm its influence on collagen 1? Many different types of collagen exist in the skin.
11. Figure 4, the significant differences should be shown in Figure 4.
12. What was reversed by L-arginine from Picrosirius red staining images? More details should be added. In another word, what did we get useful information from this result?
Comments on the Quality of English LanguageThe quality of English language should be improved.
Round 2
Reviewer 2 Report
Comments and Suggestions for Authors
The authors have replied all of my concerns. It can be accepted after checking spelling and editing problem.
Comments on the Quality of English LanguageThe description is clear.
Author Response
Dear Sir/Madam,
The manuscript has been reviewed by a proofreading company. All changes are highlighted.
Best regards,
Mihailo Rabasovic
